# Extraintestinal *Escherichia coli* from camel carcasses: Phylogroups, serotypes, and markers of virulence

**Matěj Hrala**[1], **Marina Joseph**[2], **Martina Florianová**[3], **Helena Juřicová**[3], **Ulrich Wernery**[2], **David Šmajs**[1], **Juraj Bosák**[1]*

1 Department of Biology, Faculty of Medicine, Masaryk University, Brno, Czech Republic, 2 Central Veterinary Research Laboratory, Dubai, United Arab Emirates, 3 Veterinary Research Institute, Brno, Czech Republic

* jbosak@med.muni.cz

## Abstract

Pathogenic *Escherichia coli* causes infections responsible for economic losses in animal herds worldwide. Although this bacterium is well studied in livestock and poultry, studies of camelid infections caused by extraintestinal pathogenic *E. coli* (ExPEC) are limited. In this study, a set of ExPEC from camel carcasses (n = 150) was characterized with respect to phylogenetic groups, 162 O serotypes, and 35 virulence-associated genes (VAGs) using PCR screening. ExPEC frequently belonged to phylogroup B1 (58.7%), followed by phylogroups C, A, and B2 (12.7%, 12.0%, and 9.3%, respectively). Additionally, the set of ExPEC contained 36 different serotypes. The ExPEC isolates were found to typically encode ≥5 tested VAGs, particularly those related to adhesion (*afaI*, *fimA*, *pap*, *sfa*, *tsh*), iron acquisition (*fyuA*, *iroN*, *iucC*, *sitA*), host cell damage (*α-hly*, *cdt*, *cnf1*, *sat*), invasion (*ibeA*), and bacterial protection (*iss*, *ompT*, *traT*). Moreover, ExPEC from camel adults and calves were different from each other. Among isolates from calves, prevalence was significantly higher for phylogroups C (q < 0.001) and E (q < 0.01), and ten VAGs, including fitness factors (*eitA*, *fepC*, *fyuA*, *iroN*, *iss*, *iucD*, *ompT*, *sitA*), as well as VAGs with stronger link to pathogenicity (*hlyF*, and *pap*). To identify potential reservoir of camel ExPEC strains, fecal *E. coli* (n = 139) from healthy camels were also analyzed. Based on the identified characteristics, ExPEC were distinguishable from fecal isolates of healthy camels, suggesting an exogenous source of ExPEC infections, likely transmitted from wild birds and human keepers.

## Introduction

Camel husbandry is an ecologically and economically important form of livestock farming, producing camel milk and meat [1] and utilizing arid lands that would be otherwise unsuitable for other types of farming [2]. However, commercial husbandry

**Data availability statement:** All relevant data are within the manuscript and its Supporting Information files.

**Funding:** The work was funded by the National Institute of Virology and Bacteriology (Programme EXCELES, ID Project No. LX22NPO5103, Funded by the European Union - Next Generation EU to D.S.) and by the Ministry of Health of the Czech Republic (NW25-09-00323 to J.B.). The funders had no role in study design, data collection and analysis, decision to publish, or preparation of the manuscript."

**Competing interests:** The authors have declared that no competing interests exist.

poses a high risk of animal infections [3], which lowers the economic success of camel production due to growth retardation, weight loss, and high mortality, mainly in young animals. Camel herds suffer from infections caused by various pathogens, including pathogenic *Escherichia coli* strains [4–6].

*E. coli* is a commensal bacterium colonizing human and animal intestines. At the same time, some *E. coli* strains can infect extraintestinal tissues of the host body [7]. Extraintestinal pathogenic *E. coli* (ExPEC) can cause infections ranging from common cystitis to life-threatening septicemia. ExPEC strains encode various combinations of virulence factors, which increase their virulence and ability to survive in extraintestinal environments, such as the synthesis of fimbriae, production of siderophores, and immune escape mechanisms [8]. Avian pathogenic *E. coli* (APEC) is one of the most-studied ExPEC pathotype, which primarily infects birds and poultry but also carries virulence determinants similar to human ExPEC, giving it zoonotic potential [9]. Birds can therefore act as a natural reservoir of APEC which are potentially transferable to humans and other mammals through food [10,11]. In camel husbandry, *E. coli* is responsible for diarrheal conditions [12] and may also cause camelid uterine infections, resulting in spontaneous abortions and increased neonatal mortality [5].

Characterization of pathogenic strains is critical to ensure animal welfare and the safety of animal products. The occurrence and prevalence of virulence characteristics in ExPEC strains have been extensively investigated in humans [13,14] as well as in animals in the breeding industry [15–17], but little is known about ExPEC strains causing infections in camels. Among the available tools, serotyping, particularly of the O-antigen, is a classical and widely used method to differentiate *E. coli* lineages. Specific O-serotypes are associated with particular disease conditions and epidemiology, enabling tracking, and identification of high-risk lineages [18,19]. Although camel-associated *E. coli* is largely unexplored, serotyping allows comparison with ExPEC from other hosts and may reveal potential reservoirs of infection.

This study identified and characterized the prevalence of phylogroups, O-serotypes, and virulence-associated genes (VAGs) in clinically important *E. coli* isolates taken from extraintestinal locations of camel carcasses. In addition, we found that ExPEC isolates differed in several characteristics between isolates from adult camels and their calves. We also characterized fecal *E. coli* isolates from healthy camels, assessing their role as potential reservoirs of camelid ExPEC infections.

## Materials and methods

### Collection of veterinary samples and isolation of *E. coli* strains

All veterinary samples (n = 289) were isolated from one-humped camels (*Camelus dromedarius*) and came from the commercial camel farm situated near Dubai (United Arab Emirates); the animals were raised for milk production. Animals were kept in proper open shaded pens, had access to water, and were fed high-quality feed. Calves were not separated from their mothers and had constant access to colostrum. Animals had contact with milkers twice daily and were also in contact with small animals living in this area, including birds such as pigeons.

E. coli isolated from camel carcasses (n = 150) were collected between 2004 and 2019. Isolates were considered pathogenic since they originated from extraintestinal locations (e.g., lungs, liver, brain, lymph nodes, spleen, udder, uterus, heart, kidney, and perirenal abscess). Based on the weight of camel carcasses, pathogenic E. coli isolates were described as being from adult camels (> 150 kg; n = 76) or calves (< 150 kg; n = 74). Animal carcasses were necropsied and sampled by pathologists at the Central Veterinary Research Laboratory (CVRL, Dubai) no later than 12 hours after death. Organ samples were spread on Brilliant-Green Phenol-Red Lactose Sucrose (BPLS) Agar (Merck, Germany) and aerobically cultivated for 24 hours at 37°C. Suspect lac+ colonies were further characterized using routine diagnostic methods API 20 E and a semiautomated Vitek 2 Compact system (bioMerieux, France). One isolate per sample identified as E. coli, most likely representing the dominant strain, was then preserved as a cryostock in CVRL for subsequent analysis. Fecal isolates from healthy animals (n = 139) were collected in 2020 from the same farm. Swabs were taken from the anus of adult camels with no signs of disease and were transported in Amies transport media (Deltalab, Spain) to CVRL. E. coli isolates were phenotypically identified using the same methods as for pathogenic E. coli from carcasses and one isolate per sample was then preserved as a cryostock in CVRL for subsequent analysis.

## O serotype determination of E. coli isolates

O serotyping (n = 162) was performed according to the protocol published by Iguchi et al. [20] with few modifications. Briefly, all isolates were screened using colony PCR with 20 different multiplexes containing 162 primer pairs. Each PCR reaction (20 µl) contained 200 µM deoxynucleotide triphosphates (dNTPs), 1 × ThermoPol Reaction buffer, 0.02 U/µl Taq polymerase (all from New England BioLabs, USA), and a multiplex primer mixture (2 µl) prepared according to Iguchi et al. [20]. Bacteria in water (1 µl; one colony resuspended in 200 µl of distilled water) was used as a template. The reaction mixture was supplemented with PCR-grade water (Thermo Fisher Scientific, USA). PCR amplification was performed under the following conditions: 94°C (5 min), then 35 cycles of 94°C (30 s), 55°C (30 s), and 72°C (1 min), and a final extension at 72°C (7 min). Subsequently, serotypes were determined based on PCR product size. In some cases, only the serotype group of sequentially similar serotypes was determined. In our analyses, the following serotype groups were detected: Gp2 (O28ac/O42), Gp3 (O118/O151), Gp5 (O123/O186), Gp7 (O2/O50), Gp9 (O17/O44/O73/O77/O106), Gp10 (O13/O129/O135), Gp12 (O18ab/O18ac), and Gp15 (O89/O101/O162) [for more information see Iguchi et al. [20]]. Control strains are listed in supplementary S1 Table.

## Phylogenetic determination of E. coli isolates

Phylotyping of E. coli was performed according to the protocol published by Clermont et al. [21]. Using multiplex-PCR, each E. coli isolate was classified into one of eight phylogenetic groups (A, B1, B2, C, D, E, F, or cryptic clade V). Each PCR reaction (20 µl) contained 200 µM deoxynucleotide triphosphates (dNTPs), 1 × ThermoPol Reaction buffer, 0.02 U/µl Taq polymerase (all from New England BioLabs, USA), and 1 µM of each primer. Bacteria in water were used as a template (1 µl; one colony resuspended in 200 µl of distilled water). The reaction mixture was supplemented with PCR-grade water (Thermo Fisher Scientific, USA). PCR amplification was performed under the following conditions: 94°C (5 min), then 35 cycles of 94°C (30 s), 55°C (30 s), and 72°C (1 min), and a final extension at 72°C (7 min). Appropriate E. coli strains were used as positive controls [22].

## Detection of E. coli virulence-associated genes (VAGs)

Detection of 35 virulence-associated genes (VAGs) relevant to intestinal and extraintestinal pathogenic E. coli was performed using multiplex PCR as described previously [23]. The screened VAGs were associated with binding to host cells (afaI, bfpA, eaeA, fimA, pap, pCVD432, sfa, and tsh), with iron acquisition (eitA, etsA, fepC, fyuA, ireA, iroN, iucD, and sitA), with cell and tissue damage (α-hly, cdt, cnf1, ehly, lt, pks, sat, st, stx1, stx2, and usp), with cell invasion (ibeA, ial,

and *ipaH*), or with the protection of bacterial cells (*iss*, *hlyF*, *kpsMTII*, *ompT*, and *traT*). The complete set of screened VAGs was as follows: *α-hly* – α-hemolysin, *afaI* – afimbrial adhesin, *bfpA* – bundle-forming pilus, *cdt* – cytolethal distend-ing toxin, *cnf1* – cytotoxic necrotizing factor, *eaeA* – intimin, *ehly* – enterohemolysin, *eitA* – iron transport, *etsA* – transport system, *fepC* – enterobactin transport, *fimA* – fimbriae type I, *fyuA* – yersiniabactin receptor, *hlyF* – hemolysin F, *ial* – locus associated with invasivity, *ibeA* – invasion of brain epithelium protein A, *ipaH* – locus associated with invasivity, *ireA* – iron responsive element, *iroN* – salmochelin receptor, *iss* – increased serum survival protein, *iucD* – aerobactin syn-thesis, *kpsMTII* – capsule synthesis, *lt* – thermolabile enterotoxin, *ompT* – outer membrane protease T, *pap* – P-fimbriae, pCVD432 gene/s –aggregative adherence plasmid, *pks* – colibactin, *sfa* – S-fimbriae, *sat* – secreted autotransporter toxin, *sitA* – iron transport, *st* – thermostable enterotoxin, *stx1* – Shiga toxin 1, *stx2* – Shiga toxin 2, *traT* – complement resis-tance protein, *tsh* – temperature-sensitive hemagglutinin, and *usp* – uropathogenic-specific protein. The colony PCR was performed as described above for phylotyping, using the specific primers and cycling conditions described in S2 Table. Appropriate *E. coli* positive control strains harboring individual VAG were used.

To assess the ExPEC potential of camel pathogenic *E. coli* isolates, we analyzed a set of 17 VAGs commonly linked to ExPEC strains (based on the reviews by Sarowska *et al.* [24] and Desvaux *et al.* [8]). These VAGs included: *α-hly*, *afaI*, *cdt*, *cnf1*, *fimA*, *fyuA*, *ibeA*, *iroN*, *iss*, *iucD*, *ompT*, *pap*, *sat*, *sfa*, *sitA*, *traT*, and *tsh*. For each isolate, an ExPEC-VAG score was calculated as the total number of detected ExPEC-specific VAGs. This approach allowed us to quantify the ExPEC potential.

### Statistical analysis

A two-tailed *Fisher's exact* test was used for statistical analysis of the prevalence of genetic determinants. P-values lower than 0.05 were considered statistically significant and are denoted with asterisks according to statistical significance (*$p < 0.05$, **$p < 0.01$, and ***$p < 0.001$). In cases of multiple testing, statistical significance was adjusted using the Benja-mini–Hochberg false discovery rate (FDR) procedure, controlling the FDR at 5% (*$q < 0.05$, **$q < 0.01$, and ***$q < 0.001$). GraphPad Prism 10 software was used for calculations. Correlation (i.e., correspondence analysis) was performed using R software (v4.2.0) [25].

### Ethics statement

The authors declare that the study was carried out in compliance with the ARRIVE guidelines. All *E. coli* isolates were obtained from the CVRL laboratory collection, and no farm owner permission was required.

## Results

### Collection of camel *E. coli* isolates and design of the study

A set of pathogenic *E. coli* from camel carcasses (n = 150) was collected between 2004 and 2019, and isolates were equally distributed over the 16-year period (Fig 1A). *E. coli* originated from 10 extraintestinal locations; most isolates were from the lungs and liver (44% and 36%, respectively; Fig 1B). At the same time, *E. coli* originated equally from carcasses of adults and calves (n = 76 and 74, respectively; Fig 1C). This set was furthermore characterized with respect to the prevalence of O serotypes and phylogroups. A comprehensive set of thirty-five virulence factors relevant to diarrheal and extraintestinal *E. coli* infections were also analyzed.

In addition to characterization of camel ExPEC, a set of 139 fecal *E. coli* isolates from healthy adult camels (n = 139) was collected from the same farm and analyzed as a potential source of extraintestinal infection of camels.

### Characterization of pathogenic isolates

Pathogenic *E. coli* most often belonged to phylogroup B1 (58.7%), followed by phylogroups C, A, and B2 (12.7%, 12%, and 9.3%, respectively). The three remaining phylogroups were rarely found (< 5.5%; Fig 1D). Using serotyping for 162

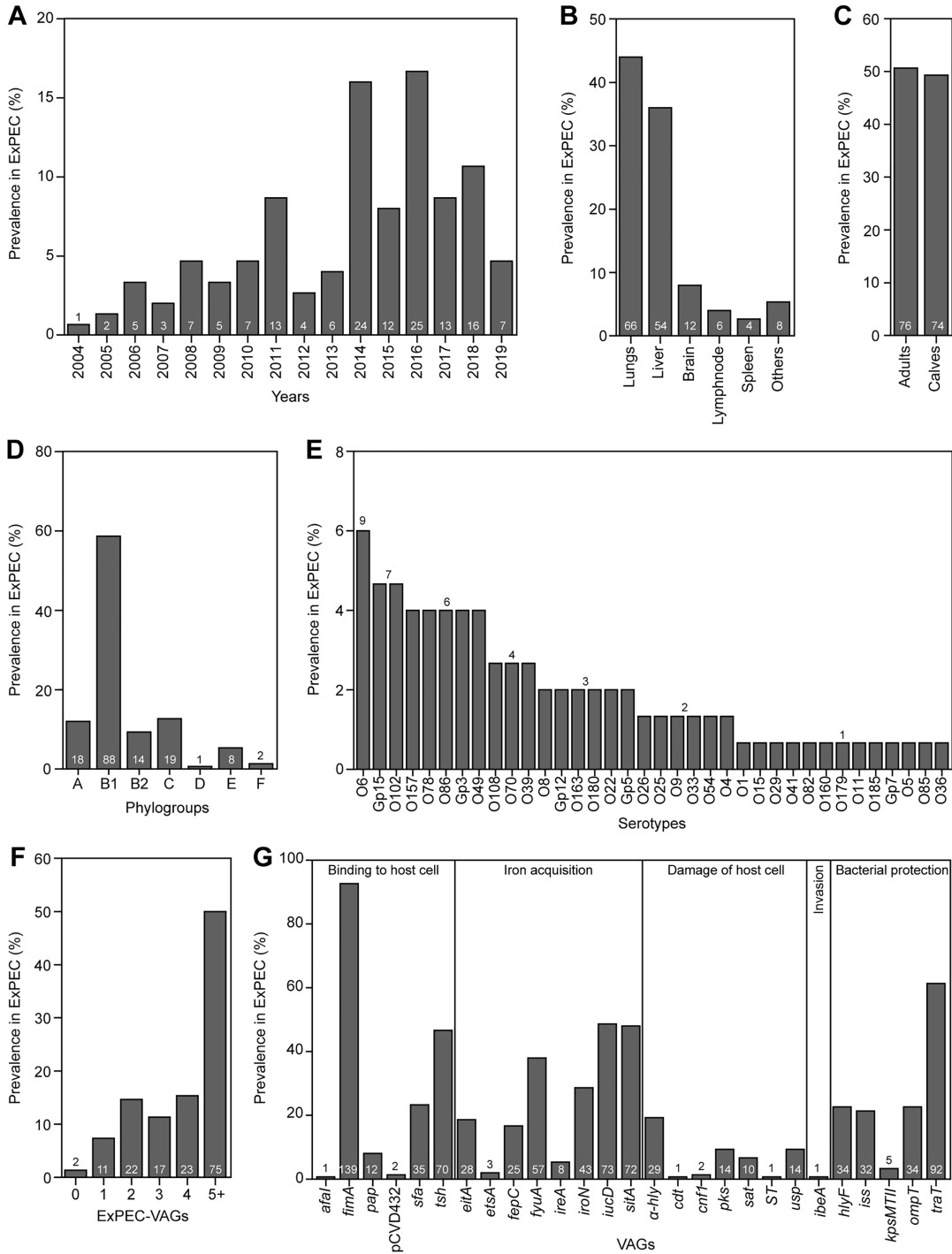

**Fig 1. Characterization of ExPEC isolates (n = 150).** Isolates were sampled over a 16-year period **(A)** and originated mainly from the lungs and liver (other locations included udders, uteruses, hearts, kidneys, and perirenal abscess) **(B)**, from adults and calves **(C)**. Camel ExPEC belongs predominantly to phylogroup B1 **(D)**. Thirty-six serotypes were detected in camel ExPEC, with serotype O6 being the most prevalent **(E)**. The majority of isolates encoded several VAGs previously associated with ExPEC [8,24] **(F)**. Twenty-seven different VAGs were detected in the ExPEC set, which included a high prevalence of genes encoding for adhesion, iron acquisition, and bacterial protection **(G)**. The numbers displayed in or above each column indicate the number of isolates. A complete list of detected characteristics is shown in S3 Table.

different O serotypes or groups of serotypes, 72% of pathogenic *E. coli* were successfully classified (n = 108); the rest of the isolates were non-typeable. Thirty-six serotypes or serogroups were found among the pathogenic *E. coli*. Serotypes O6, Gp15 (O89/101/162), and O102 were the most prevalent (6.0%, 4.7%, and 4.7%, respectively). Moreover, serotype O157 was found in six isolates (4.0%). The non-typeable isolates (n = 42) included isolates from various times, sources, and phylogroups, suggesting a non-clonal character (S3 Table).

The presence of 35 virulence-associated genes (VAGs) was further analyzed. Specifically, *E. coli* were screened for eight specific VAGs associated with five important diarrheal pathotypes (EPEC (*eaeA*), ETEC (*st*, *lt*), STEC (*stx1*, *stx2*), EIEC (*ial*, *ipaH*), and DAEC (*afaI*) [26,27]. Even though diseased camels frequently suffered from diarrhea before death (personal observation; U. Wernery), diarrheal *E. coli* was not found in our set, except for one ETEC (*st*+) and one DAEC (*afaI*+) isolate (S3 Table). Since ExPEC are a heterogeneous group of strains frequently harboring a flexible pool of VAGs, the prevalence of 17 VAGs typical of ExPEC isolates [8,24] was assessed in our samples of camel pathogenic *E. coli* to evaluate their ExPEC potential (see Methods). On average, pathogenic *E. coli* harbored 4.6 ExPEC-VAGs, and 50% of isolates harbored five or more different ExPEC-associated VAGs (Fig 1F). At the same time, only two isolates were found not to harbor any of the tested ExPEC-VAGs. These findings confirmed the classification of isolates as ExPEC based on both the source of isolation and the presence of ExPEC genetic markers.

Regarding individual VAGs, 27 out of the 35 tested determinants were found among camel pathogenic *E. coli*. Camel pathogenic *E. coli* frequently harbored adhesins, iron acquisition systems, and protectins (Fig 1G). Besides the widely distributed fimbriae type 1 (92.7% *fimA*), we also found the following VAGs to be notably prevalent: *traT* (61.3%), *iucD* (48.7), *sitA* (48.0%), *tsh* (46.7%), *fyuA* (38.0%), *iroN* (28.7%), *sfa* (23.3%), *hlyF* (22.7%), *ompT* (22.7%), *iss* (21.3%), and *α-hly* (19.3%).

Based on the distribution of phylogroups, serotypes, and VAGs, 118 different pathogenic *E. coli* types were identified, which preclude the expansion of either a single or a few pathogenic *E. coli* clones from being responsible for camel infections. A complete list of characteristics for individual *E. coli* isolates is shown in S3 Table.

## Pathogenic *E. coli* isolates from camel adults and calves differ in their characteristics

In addition to the characterization of camel pathogenic isolates as a whole set (Fig 1), four subsets of *E. coli* representing their origin [i.e., lungs (n = 66), liver (n = 54), adults (n = 76), and calves (n = 74)] were further analyzed and compared relative to their characteristics. While no significant differences were found between ExPEC from the lungs and liver (S4 Table), several differences were found between ExPEC from adults and calves.

ExPEC from calves less frequently belonged to the B1 phylogroup compared to ExPEC isolates from adults (32.4% and 84.2%, respectively; q < 0.001); at the same time, the calves ExPEC group was significantly enriched, compared to adults, with isolates belonging to phylogroups C (24.3% vs. 1.3%; q < 0.001) and E (10.8% vs. 0.0%; q < 0.01; Fig 2A). Fifteen serotypes were found specific for ExPEC of adults and 13 serotypes were only in ExPEC of calves (S3, S5 Tables).

ExPEC isolates from adults and calves differed in 12 VAGs. While *α-hly* and *tsh* were more common among ExPEC from adult camels (q < 0.001), the prevalence of *pap*, *iucD* (q < 0.05) and eight other VAGs [*eitA*, *fyuA*, *hlyF*, *iroN*, *iss*, *ompT*, and *sitA* (q < 0.001), and *fepC* (q < 0.01)] was significantly higher in ExPEC isolates from calves (Fig 2B). Most VAGs were for various iron acquisition systems (*iucD*, *eitA*, *fepC*, *fyuA*, *iroN*, and *sitA*, prevalence 27–77%) and three different protectins (*hlyF*, *iss*, and *ompT*, prevalence 37.8–39.2%). A list of characteristics of the individual isolates is shown in S3 Table, and complete comparisons of *E. coli* subsets are shown in S5 Table.

## Fecal isolates do not represent an important source of ExPEC

To identify the reservoir of camel ExPEC, a set of fecal *E. coli* (n = 139) from healthy dromedars was collected, characterized, and compared to ExPEC.

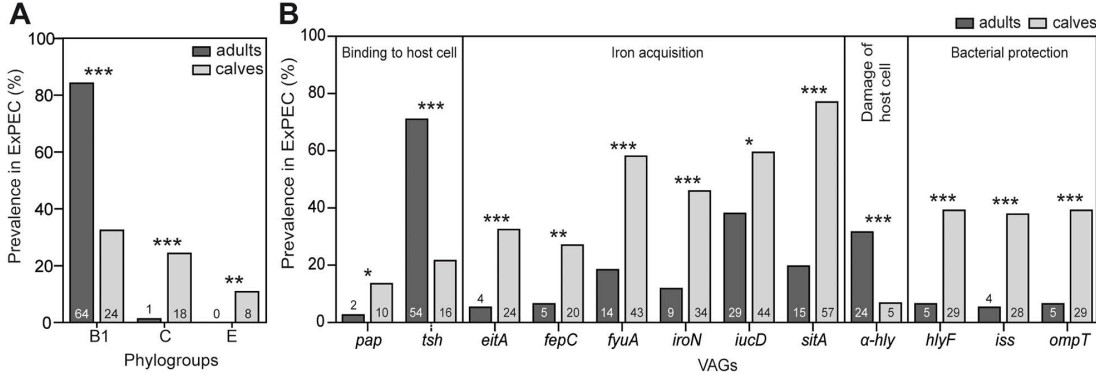

**Fig 2. Characteristics with significantly different prevalence between ExPEC from adult camels (n = 76) and calves (n = 74). A)** While the ExPEC from adults mostly belonged to phylogroup B1, the calf ExPEC contained significantly more isolates from phylogroups C and E. **B)** The prevalence of *tsh* and *α-hly* was higher in ExPEC from adult camels, while calf ExPEC frequently encoded various iron acquisition systems and bacterial protectins. The numbers displayed in or above each column indicate the respective number of isolates. The two-tailed *Fisher's exact* test was used to calculate the statistical significance, and values were adjusted for multiple testing using the Benjamini–Hochberg false discovery rate (FDR) procedure, controlling the FDR at 5% (*q < 0.05, **q < 0.01, ***q < 0.001). A complete list of detected characteristics is shown in S3 Table.

Similar to adult ExPEC, fecal *E. coli* predominantly belonged to phylogroup B1 (80.6%), while B2, D, E, and F phylogroups were found only sporadically. Half of the fecal *E. coli* (51%) were successfully serotyped and classified into 49 distinct serotypes. The most frequently identified VAGs were *fimA* (71.9%) and *traT* (54.7%). The remaining VAGs had prevalences of 5.8% or lower (Fig 3A; S3 Table).

Using correspondence analysis of all identified characteristics, fecal isolates represented a distinct group compared to ExPEC isolates (Fig 3B). In addition, different clusters were observed even between adult and calf ExPEC isolates. While calf ExPEC showed an association with phylogroups B2, C, D, E, and F, along with the most of the identified VAGs, ExPEC from adults were associated with *α-hly* and *tsh* (*cdt* and *st* were detected in just one isolate each).

In addition, despite the distinct clusters in the correspondence analysis, two identical patterns of characteristics (i.e., phylogroups, serotypes, and VAGs) were shared among all three *E. coli* groups (adult and calf ExPEC and fecal *E. coli*; Fig 3C). Altogether, out of 189 identified patterns, nine were shared between fecal *E. coli* and ExPEC, corresponding to 38 fecal but only 11 ExPEC isolates. Moreover, these isolates were represented mainly by patterns with low VAGs abundance and negative serotype detection (typically B1, „non-serotypeable", *fimA*; Fig 3C).

## Discussion

*E. coli* in camel herds causes both intestinal diarrheal [28] and extraintestinal infections, such as septicemias, which are predominantly diagnosed in neonatal calves and are usually fatal. On camel farms, the prevalence of colisepticemia can reach up to 5% (personal observation; U. Wernery). For this study, a unique and comprehensive set of clinically relevant ExPEC isolates taken from camel carcasses was isolated over almost two decades. While camel diarrheal *E. coli* has been previously characterized with respect to serotypes and encoded virulence factors [29–31], this is the first study characterizing camel ExPEC. Moreover, camel fecal *E. coli* was analyzed as a potential source of ExPEC infections.

In our set of pathogenic isolates (n = 150), a total of 118 different patterns of detected characteristics (i.e., phylogroups, serotypes, and VAGs) were identified (Fig 3C). Although pathogenic *E. coli* tends to cause epidemic infections involving the prevailing clones [32,33], here we found notable diversity among the pathogenic isolates. The set of pathogenic *E. coli* from camel carcasses was isolated over a 16-year period, and during the period, no spread of any *E. coli* clones

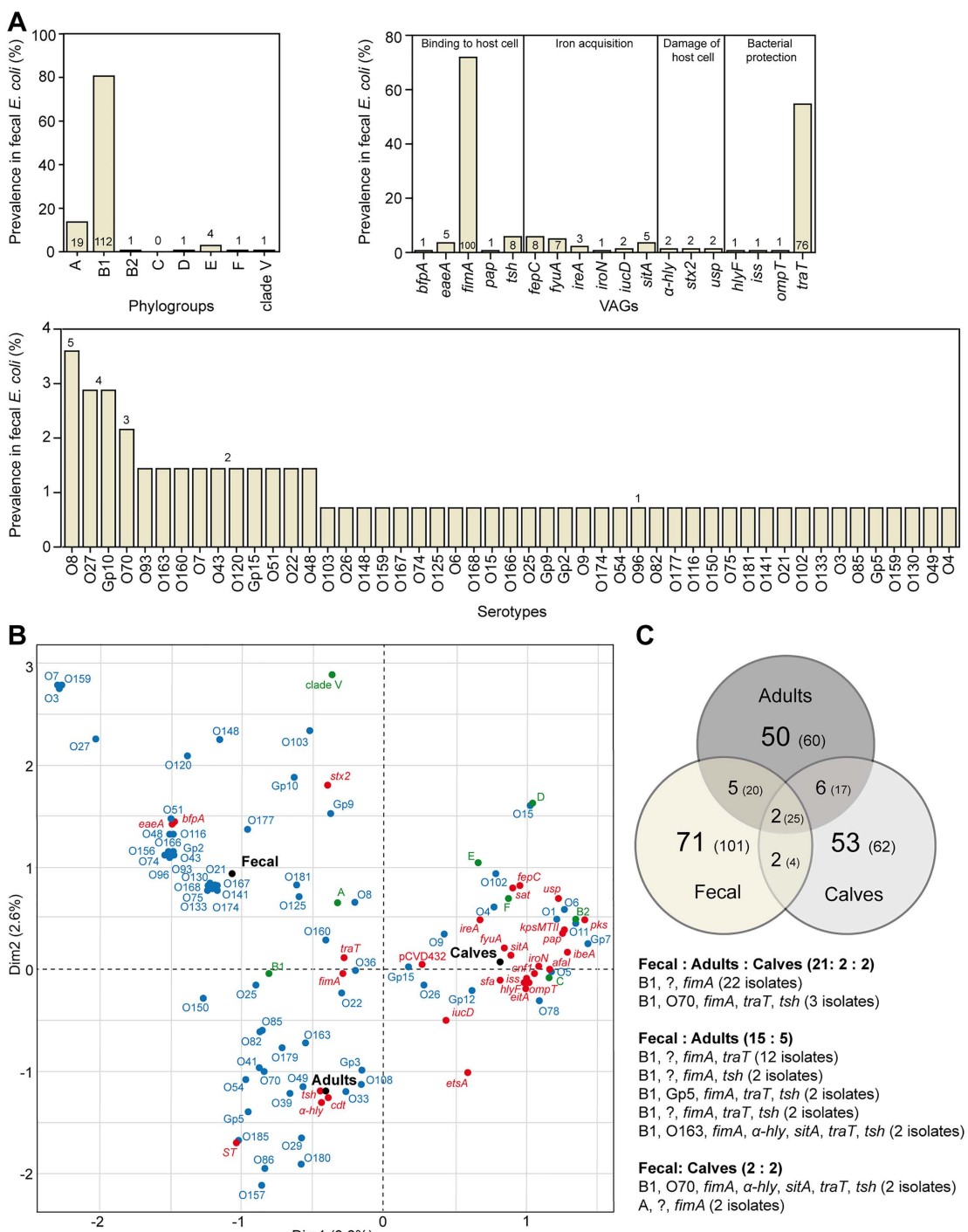

**Fig 3. Characterization of fecal *E. coli* from healthy adult camels (n = 139) and comparison with the ExPEC set. A)** Phylogroup B1 was prevalent among fecal isolates from healthy camels. Except for *fimA* and *traT*, other VAGs were detected at low frequencies. A high diversity in fecal *E. coli* serotypes was found. The numbers displayed in or above each column indicate the respective number of isolates. **B)** Correspondence analysis of all camel *E. coli* isolates and their characteristics revealed three distinct clusters, reflecting the origins of *E. coli*. *E. coli* source is highlighted in black, phylogroups are shown in green, serogroups in blue, and VAGs in red. **C)** The Venn diagram shows each set's unique and shared *E. coli* patterns. The numbers indicate the number of patterns found, while the numbers in brackets represent the number of isolates. The list of patterns shared between fecal *E. coli* and ExPEC is shown below the Venn diagram. Each row contains a distinct pattern, indicating the present phylogroup, serotype (with non-serotypeable patterns denoted by "?"), and VAGs.

was evident (S3 Table). Therefore, given the variability in these patterns and the duration of sample collection, the set represent a comprehensive and clinically relevant isolates without clonal characteristics.

Pathogenic *E. coli* was mostly isolated from internal organs, such as the lungs and liver. Defining ExPEC based on observed characteristics can be challenging, therefore we used a recently published list of ExPEC-VAGs [8,24] to verify the ExPEC character of our isolates. Half of our isolates harbored five or more ExPEC-VAGs (only two isolates did not harbor any of these, Fig 1F), further supporting their potential to cause extra-intestinal infection. In addition to ExPEC-VAGs, two isolates harbored genes typical of intestinal-pathogenic *E. coli* (i.e., ETEC and DAEC).

In our set, the most prevalent phylogroup was B1 (58.6%), followed by phylogroups C, A, B2, and E; phylogroups D and F were found only rarely. Phylogroup B1 was also frequently found (51–60%) in *E. coli* isolated from cattle [34,35], which could reflect the nutritional relatedness of cattle and camels. Phylogroup B1 was also the most prevalent group in a large-scale study of porcine ExPEC (54%; [36]). Additionally, pig septicemia isolates primarily belonged to groups A and B1 [37]. As with our finding, Bessalah *et al.* [29] found a higher prevalence of phylogroups B2 and C among pathogenic *E. coli* from diarrheal camels, and Louge Uriarte *et al.* [38] found that phylogroup C was the most prevalent among *E. coli* from dairy calves with septicemia and meningitis.

Phylogroup B1 is uncommon among human *E. coli* from healthy persons or patients with a variety of conditions [23,39,40] and these isolates are considered to be transient human colonizers. However, a phylogroup B1 lineage responsible for human extra-intestinal infections and antibiotic resistance has recently emerged [17,41]. While phylogroup B2 was rarely found in our isolates of fecal *E. coli* (0.7%), it was common among our ExPEC isolates (9.3%). *E. coli* in the B2 phylogroup are long-term human colonizers rarely found in veterinary isolates [22,42]. Moreover, phylogroup B2 is typical for human ExPEC infections [43–47]. The presence of the typical human B2 phylogroup in camel ExPEC supports the hypothesis that these strains are transmitted to camels from their keepers. Interestingly, we found that phylogroup C was common and specific for calf ExPEC and rare or completely absent among isolates from diseased and healthy adult camels. Phylogroup C was also found to be associated with ExPEC, which is responsible for human urinary infections [48].

Characterization of variations in the O antigen is a standard technique for epidemiological analyses and tracing of infection sources [19,49]. In this study, we analyzed the prevalence of the majority of known serotypes (i.e., 162 of 187 that have been defined; [19,20]). Despite the broad set of tested serotypes, 28.0% of camel ExPEC isolates remained unserotyped, which may indicate that a considerable number of camel *E. coli* serotypes may still be unknown. The most prevalent serotype among camel ExPEC was O6 (6%). This serotype has been also detected in intestinal *E. coli* from camels in Tunisia [29] and in extraintestinal infections of humans and birds [50,51]. Moreover, we detected six ExPEC isolates with the O157 serotype (4%), and interestingly, all were Shiga toxin-negative (*stx*-). Diarrheal O157 Stx-negative strains have been previously isolated from cattle, pigs, and humans [52,53] and have been shown to still cause diarrheal infections. A similar prevalence of *E. coli* O157 was found in diarrheal camels in Tunisia (3.3% [54]) and the United Arab Emirates (4.3% [55]). While *E. coli* O157:H7 is a well-characterized zoonotic intestinal pathogen [56,57], our study suggests that *E. coli* O157 can also cause sporadic extraintestinal infections in camels.

The most prevalent VAGs among ExPEC isolates were *fimA*, *iucD*, *sitA*, *traT,* and *tsh*. Since *fimA* and *traT* showed widespread distribution in both fecal and pathogenic camel *E. coli*, these VAGs are unlikely to be significant contributors to camel ExPEC infections. Such broadly distributed traits are generally considered fitness factors [58], as they enhance bacterial persistence and colonization but are not necessarily linked to high virulence unlike other traits [59,60]. At the same time, camel ExPEC isolates frequently encoded *iucD* and *sitA*. Presence of these two iron acquisition systems (aerobactin and ferrous iron/manganese uptake) supports their importance relative to human and animal ExPEC [24,45,61–64]. Iron uptake systems are regarded as important virulence determinants as they provide pathogens with advantage in the iron-limited host environment, such as in the extraintestinal environment and during inflammation [65]. Temperature-sensitive hemagglutinin (*tsh*) is a virulence factor contributing to the successful adherence of bacteria to eukaryotic cells [9] and has previously been associated with avian-pathogenic *E. coli* (APEC) [66,67]. Moreover, Ovi *et al.*

[67] identified several key APEC VAGs, some of which were also associated with the camel ExPEC in our study (i.e., *hlyF*, *iroN*, *iss*, *iucD*, *ompT*, *pap*, and *tsh*). Since our camel ExPEC strains share significant genetic and pathogenic similarities with APEC strains [11], the avian origin of camel infections appears possible since camels are kept in open corrals that allow contact with wild animals, including birds.

In addition to potential human/bird-to-camel transfer scenarios, we experimentally investigated the fecal *E. coli* in healthy camels as a potential source of calf infections. Correlation analyses showed a clear differentiation between ExPEC and fecal *E. coli* isolates from healthy camels. Moreover, among patterns of detected characteristics, only a small overlap between fecal and ExPEC isolates was found, and the majority of these overlaps were from the fecal *E. coli* set (Fig 3C). These overlapping isolates encoded a small number of VAGs (mainly represented by *fimA* and *traT*). Based on these results, we hypothesize that healthy adult camels do not represent an important reservoir of camel ExPEC infections. On the other hand, Bessalah *et al.* [29] frequently found *iroN*, *iss*, *iucD*, *kpsMTII,* and *pap* in a set of seventeen *E. coli* from healthy camels in Tunisia.

The VAGs identified in this study are usually part of virulence plasmids or pathogenicity islands, which facilitate their horizontal transfer among *E. coli* isolates [8,64]. For example, VAGs associated with infections of calves, such as *hlyF*, *iss*, *iucD*, *ompT*, and *sitA* are typically encoded on plasmid ColV, which is a well-known contributor to the pathogenicity of ExPEC [17]. Interestingly, this plasmid was associated with poultry and porcine but not bovine sources [17].

A limitation of this study is that only one *E. coli* isolate per sample, representing the potentially dominant strain, was subjected to genetic characterization. In the future, genome sequencing on this set of isolates would provide deeper insight into camel ExPEC, especially among isolates having a lower prevalence of the tested characteristics.

## Conclusion

In this study, we characterized the extensive set of extraintestinal pathogenic *Escherichia coli* isolated from camels. Based on our data, we hypothesize that the source of camel ExPEC infections is exogenous, coming from either wild birds or human keepers, although further studies are needed to confirm these transmission routes. This information could help prevent camel farm infections.

## Supporting information

**S1 Table. List of analyzed serotypes (Iguchi *et al.* 2015) and control strains.**
(XLSX)

**S2 Table. List of primers and cycling conditions used for detection of VAGs.**
(XLSX)

**S3 Table. List of isolates and their characteristics.**
(XLSX)

**S4 Table. Comparison of isolates originating from livers and lungs among camel ExPEC (n = 150).**
(XLSX)

**S5 Table. Comparison of analyzed characteristics between ExPEC from adults and calves.**
(XLSX)

## Acknowledgments

We thank Thomas Secrest (Secrest Editing, Ltd.) for his assistance with the English revision of the manuscript.

## Author contributions

**Conceptualization:** Ulrich Wernery, David Šmajs, Juraj Bosák.

**Data curation:** Matěj Hrala, Juraj Bosák.

**Formal analysis:** Matěj Hrala.

**Funding acquisition:** David Šmajs, Juraj Bosák.

**Methodology:** Matěj Hrala, Marina Joseph, Martina Florianová, Helena Juřicová.

**Project administration:** Helena Juřicová, David Šmajs.

**Resources:** Marina Joseph, Martina Florianová, Helena Juřicová, Ulrich Wernery.

**Supervision:** Ulrich Wernery, David Šmajs, Juraj Bosák.

**Validation:** Helena Juřicová.

**Visualization:** Matěj Hrala.

**Writing – original draft:** Matěj Hrala.

**Writing – review & editing:** Marina Joseph, Martina Florianová, Helena Juřicová, Ulrich Wernery, David Šmajs, Juraj Bosák.

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
