## [Decision Letter · Decision Letter 0]

23 Jul 2025

Dear Dr. Bosák,

Thank you for submitting your manuscript to PLOS ONE. After careful consideration, we feel that it has merit but does not fully meet PLOS ONE’s publication criteria as it currently stands. Therefore, we invite you to submit a revised version of the manuscript that addresses the points raised during the review process.

The manuscript provides a comprehensive study on extraintestinal pathogenic Escherichia coli (ExPEC) from camel carcasses, focusing on phylogroups, serotypes, and virulence-associated genes (VAGs). Reviewers commend its clarity and organization but highlight critical areas needing improvement in methodology, data interpretation, and discussion.

Additional Editorial Requirements:

1. Ensure all data underlying findings are fully accessible, clearly stating where data is deposited or how it can be accessed. Include a Data Availability Statement and justify any restrictions per PLOS ONE’s policy.

2. Provide a detailed description of statistical methods used, including assumptions and handling of multiple comparisons (e.g., Bonferroni correction), clearly linking tests to results.

3. Revise the Abstract and Discussion for clarity, particularly regarding the significance of VAGs and implications of NT isolates, ensuring polished language throughout.

4. Confirm the ethics statement accuracy, stating ethical approval details for vertebrate animal involvement, or clarify if no approval was required for existing laboratory collections.

We look forward to receiving your revised manuscript.

Kind regards,

Nabi Jomehzadeh, Ph.D (Assistant Professor)

Academic Editor

PLOS ONE

Journal Requirements:

“The work was funded by the National Institute of Virology and Bacteriology (Programme EXCELES, ID Project No. LX22NPO5103, Funded by the European Union - Next Generation EU) to DS.”

4. Please include captions for your Supporting Information files at the end of your manuscript, and update any in-text citations to match accordingly. Please see our Supporting Information guidelines for more information: http://journals.plos.org/plosone/s/supporting-information .

Reviewers' comments:

Reviewer's Responses to Questions

**Comments to the Author**

1. Is the manuscript technically sound, and do the data support the conclusions?

Reviewer #1: Yes

Reviewer #2: No

2. Has the statistical analysis been performed appropriately and rigorously?

Reviewer #1: I Don't Know

Reviewer #2: N/A

3. Have the authors made all data underlying the findings in their manuscript fully available?

Reviewer #1: Yes

Reviewer #2: Yes

4. Is the manuscript presented in an intelligible fashion and written in standard English?

Reviewer #1: Yes

Reviewer #2: Yes

Reviewer #1: Reviewer's report

PONE-D-25-05895: Extraintestinal Escherichia coli from camel carcasses: Phylogroups, serotypes, and markers of virulence.

Escherichia coli has received increasing attention over the past decade as a cause of various intestinal and extraintestinal infections in humans and animals.

Infection with E. coli has serious consequences, such as growth retardation, weight loss and high mortality in farm animals, especially young animals. It is a major economic problem.

Extraintestinal pathogenic E. coli isolated from camel carcasses and fecal E. coli from healthy adult camels (as controls) were characterised by the authors of this manuscript for phylogenetic groups, O-serotypes and virulence-associated genes. Isolates from adult camels and young individuals from the same farm were compared. The researchers attempted to identify the source and transmission route of the bacteria by comparing fecal isolates from healthy camels as potential reservoirs of ExPEC infections.

The manuscript is written clearly and well organized but requires corrections.

Comments

In the Abstract and Discussion, there is a lack of opinion on the character of virulence factors. What is the significance of the virulence factors found? Not only the quantity but also the type of Vf is important, some belong to the so-called fitness factors, others indicate high virulence of the strains. The general role of Vf is tabulated, but this requires discussion.

The Introduction should describe the pathotypes of E. coli, particularly APEC, as the conclusion included information that APEC strains from wild birds may be responsible for infections.

Materials and methods

Details of the methodology sufficient to allow the experiments to be reproduced, but:

1. Only one isolate per sample was genetically tested, but I have doubts about whether this is a good approach. There may be several different E. coli genotypes with different genetic characteristics in one sample. At least 3-5 isolates with # phenotypes should be analysed. Clonal relationships can be excluded by genetic typing (e.g. PCR fingerprinting). Explanation is required.

2. In PCR, final concentrations in the reaction mixture should be reported, not volumes.

3. I understand that the same matrix was used in determining the phylogenetic group, serological group or Vf, there is no need to describe the method of DNA isolation for each PCR (L: 104 – 106; L: 121-123; L: 137-139)

4. L: 148 - pCVD432 is a large plasmid, according to me the information should be: pCVD432 gene/s

Results/ Discussion

1. L: 164-168 This information can be omitted, it is a repeat from Materials and Methods.

2. As a result of the research, it was found that the most common phylogenetic group was B1. This is contrary to the research results available in many scientific studies. The most pathogenic is group B2 and D. B1 belongs to the commensal group. It is worth discussing.

3. The significance of serotypes should be discussed. What do they indicate?

4. L: 199-200 I don't understand this information: "Desvaux et al., [23]) was assessed in our samples of camel pathogenic E. coli."

5. The discussion on virulence factors should be more specific. What are the consequences of the presence of these genes and what role could they play in pathogenesis? Only statistics are given without indicating which Vf are important.

Reviewer #2: Dear researchers, thank you for selecting this intriguing topic. However, the research requires a general revision, which is hereby announced. Dear researchers, thank you for selecting this intriguing topic. However, the research requires a general revision, which is hereby announced. It has come to our attention that certain aspects of the methodology lack sufficient clarity, and the data analysis section requires a more rigorous approach.

We kindly urge you to revisit the research design and ensure that the hypotheses are clearly defined and adequately supported by the literature. Furthermore, the conclusions drawn need to be more closely aligned with the data presented, avoiding any overgeneralizations.

**Do you want your identity to be public for this peer review?** For information about this choice, including consent withdrawal, please see our Privacy Policy

Reviewer #1: **Yes: ** Beata Krawczyk

Reviewer #2: No

---

## [Author Response · Author response to Decision Letter 1]

8 Sep 2025

PONE-D-25-05895

Extraintestinal Escherichia coli from camel carcasses: Phylogroups, serotypes, and markers of virulence

PLOS ONE

Editor´s comments

The manuscript provides a comprehensive study on extraintestinal pathogenic Escherichia coli (ExPEC) from camel carcasses, focusing on phylogroups, serotypes, and virulence-associated genes (VAGs). Reviewers commend its clarity and organization but highlight critical areas needing improvement in methodology, data interpretation, and discussion.

Additional Editorial Requirements:

1. Ensure all data underlying findings are fully accessible, clearly stating where data is deposited or how it can be accessed. Include a Data Availability Statement and justify any restrictions per PLOS ONE’s policy.

The Data availability statement has been added to the Methods section:

Lines 185-186: “All relevant data are within the manuscript and its supplementary material files or are available from the authors upon request.”

2. Provide a detailed description of statistical methods used, including assumptions and handling of multiple comparisons (e.g., Bonferroni correction), clearly linking tests to results.

To ensure reliability of our findings during multiple statistical tests, we decided to adjust for the false discovery rate (FDR) using the Benjamini–Hochberg procedure. Its use is now stated in the Methods section.

Methods Lines 179-180: “In cases of multiple testing, statistical significance was adjusted using the Benjamini–Hochberg false discovery rate (FDR) procedure, controlling the FDR at 5% (q < 0.05).”

Figure legend Lines 284-286: “The two-tailed Fisher’s exact test was used to calculate the statistical significance (*q < 0.05, **q < 0.01, ***q < 0.001), and values were adjusted for multiple testing using the Benjamini–Hochberg false discovery rate (FDR) procedure, controlling the FDR at 5% (q < 0.05).”

The supplementary materials (Table S4, Table S5) were also updated to reflect this correction.

The application of FDR did not change the statistical significance of the VAGs and phylogroup differences between adults and calves. However, the serotype comparison became non-significant after the adjustment. Therefore, we have removed these non-significant results from Figure 2 and deleted the corresponding text about serotype significance (Lines 37; 264-266; 279-280).

3. Revise the Abstract and Discussion for clarity, particularly regarding the significance of VAGs and implications of NT isolates, ensuring polished language throughout.

The statements clarifying the biological relevance of the detected VAGs have been added to the Abstract and Discussion section.

Lines: 37-39; 385-387; 390-393.

4. Confirm the ethics statement accuracy, stating ethical approval details for vertebrate animal involvement, or clarify if no approval was required for existing laboratory collections.

We confirm the existing statement (Lines 438-440): “The authors declare that the study was carried out in compliance with the ARRIVE guidelines. All E. coli isolates were obtained from the CVRL laboratory collection, and no farm owner permission was required.”

Journal Requirements:

“The work was funded by the National Institute of Virology and Bacteriology (Programme EXCELES, ID Project No. LX22NPO5103, Funded by the European Union - Next Generation EU) to DS.”

The new funder has been added alongside the existing one, and the roles of the funders have been specified.

Lines 447-451: “The work was funded by the National Institute of Virology and Bacteriology (Programme EXCELES, ID Project No. LX22NPO5103, Funded by the European Union - Next Generation EU) to DS and by the Ministry of Health of the Czech Republic (NW25-09-00323) to JB. The funders had no role in study design, data collection and analysis, decision to publish, or preparation of the manuscript.”

The phrase “data not shown” was removed from the manuscript. The referenced data are not a core part of the research presented.

The supplementary material captions are listed on Lines 429-436.

N/A

Reviewers' comments:

Reviewer #1: Reviewer's report

PONE-D-25-05895: Extraintestinal Escherichia coli from camel carcasses: Phylogroups, serotypes, and markers of virulence.

Escherichia coli has received increasing attention over the past decade as a cause of various intestinal and extraintestinal infections in humans and animals.

Infection with E. coli has serious consequences, such as growth retardation, weight loss and high mortality in farm animals, especially young animals. It is a major economic problem.

Extraintestinal pathogenic E. coli isolated from camel carcasses and fecal E. coli from healthy adult camels (as controls) were characterised by the authors of this manuscript for phylogenetic groups, O-serotypes and virulence-associated genes. Isolates from adult camels and young individuals from the same farm were compared. The researchers attempted to identify the source and transmission route of the bacteria by comparing fecal isolates from healthy camels as potential reservoirs of ExPEC infections.

The manuscript is written clearly and well organized but requires corrections.

Comments

In the Abstract and Discussion, there is a lack of opinion on the character of virulence factors. What is the significance of the virulence factors found? Not only the quantity but also the type of Vf is important, some belong to the so-called fitness factors, others indicate high virulence of the strains. The general role of Vf is tabulated, but this requires discussion.

We thank the reviewer for this important point. We added the biological significance of the most prevalent VFs identified in camel ExPEC throughout the manuscript:

Abstract, Lines 37-39: “…and ten VAGs, including fitness factors (eitA, fepC, fyuA,iroN, iss, iucD, ompT, sitA), as well as VAGs with stronger link to pathogenicity (hlyF, and pap).”

Discussion, Lines 385-387: “. Such broadly distributed traits are generally considered fitness factors [59], as they enhance bacterial persistence and colonization but are not necessarily linked to high virulence unlike other traits [60,61].”

Lines 390-393: “Iron uptake systems are regarded as important virulence determinants as they provide pathogens with advantage in the iron-limited host environment, such as in the extraintestinal environment and during inflammation [66].”

The Introduction should describe the pathotypes of E. coli, particularly APEC, as the conclusion included information that APEC strains from wild birds may be responsible for infections.

The Introduction section has been expanded to include information about APEC:

Lines 58-62: “Avian pathogenic E. coli (APEC) is one of the most-studied ExPEC pathotype, which primarily infects birds and poultry but also carries virulence determinants similar to human ExPEC, giving it zoonotic potential [9]. Birds can therefore act as a natural reservoir of APEC which are potentially transferable to humans and other mammals through food [10,11].”

Materials and methods

Details of the methodology sufficient to allow the experiments to be reproduced, but:

1. Only one isolate per sample was genetically tested, but I have doubts about whether this is a good approach. There may be several different E. coli genotypes with different genetic characteristics in one sample. At least 3-5 isolates with # phenotypes should be analysed. Clonal relationships can be excluded by genetic typing (e.g. PCR fingerprinting). Explanation is required.

We acknowledge the reviewer’s concern regarding the potential presence of multiple E. coli genotypes in specimen. In our study, however, we aimed to focus on the dominant isolate. Throughout the long sampling period (2014–2019), we consistently selected and cryopreserved only one colony from those having the same morphology on the agar plate for analysis. We believe that the characterization of the most dominant isolate type per sample is sufficient for this type of study. We acknowledge this limitation and have highlighted it in the Methods and Discussion.

Method Lines 100-101: “One isolate identified as E. coli per sample, most likely representing the dominant strain, was then preserved as a cryostock in CVRL for subsequent analysis.”

Lines 105-106: “…and one isolate was then preserved as a cryostock in CVRL for subsequent analysis.”

Discussion Lines 418-420: “A limitation of this study is that only one E. coli isolate per sample, representing the potentially dominant strain, was subjected to genetic characterization.”

2. In PCR, final concentrations in the reaction mixture should be reported, not volumes.

As recommended, we have edited the Methods to report the final concentrations in the reaction mixture instead of the volumes.

Lines 113-115: “Each PCR reaction (20 µl) contained 200 µM deoxynucleotide triphosphates (dNTPs), 1×ThermoPol Reaction buffer, 0.02 U/µl Taq polymerase (all from New England BioLabs, USA), and a multiplex primer mixture (2 µl) prepared according to Iguchi et al. (2015).”

Lines 132-134: “Each PCR reaction (20 µl) contained 200 µM deoxynucleotide triphosphates (dNTPs), 1×ThermoPol Reaction buffer, 0.02 U/µl Taq polymerase (all from New England BioLabs, USA), and 1 µM of each primer.”

3. I understand that the same matrix was used in determining the phylogenetic group, serological group or Vf, there is no need to describe the method of DNA isolation for each PCR (L: 104 – 106; L: 121-123; L: 137-139)

We appreciate the reviewer's note for improvement. The redundancies in the Methods section have been removed.

Lines 148-149: “The colony PCR was performed as described above for phylotyping, using the specific primers listed in Table S2.”

4. L: 148 – pCVD432 is a large plasmid, according to me the information should be: pCVD432 gene/s

We appreciate the reviewer's comment. Our previous phrasing has been edited to accurately reflect this point (Line 162).

Previously: “pCVD432 – aggregative adherence plasmid”

Edited to: “pCVD432 gene/s – aggregative adherence plasmid”

Results/ Discussion

1. L: 164-168 This information can be omitted, it is a repeat from Materials and Methods.

We thank the reviewer for this point. The corresponding paragraph has been omitted to avoid redundancy.

2. As a result of the research, it was found that the most common phylogenetic group was B1. This is contrary to the research results available in many scientific studies. The most pathogenic is group B2 and D. B1 belongs to the commensal group. It is worth discussing.

We thank the reviewer for this comment. We agree with the reviewer that phylogroup B2 is the most prevalent in human ExPEC. Veterinary ExPEC isolates (pigs and cattle), on the others hand, show results similar to our own. Interestingly, some recent studies have also isolated phylogroup B1 from human bloodstream infections (Santos et al. 2023; doi: 10.1007/s42770-022-00884-1).

We have expanded the Discussion section to include this additional information.

Lines 350-352: “Phylogroup B1 was also the most prevalent group in a large-scale study of porcine ExPEC (54%; [36]). Additionally, pig septicemia isolates primarily belonged to groups A and B1 [37].”

Citation added; Lines 358-359: “However, a phylogroup B1 lineage responsible for human extra-intestinal infections and antibiotic resistance has recently emerged [17,41].”

3. The significance of serotypes should be discussed. What do they indicate?

Specific O-serotypes have been linked to particular ExPEC clones in humans and animals, enabling the identification of high-risk strains and the tracing of infection sources, which was particularly relevant in our study. However, camel ExPEC interestingly showed a high diversity in O-serotypes and could not be clearly linked to a specific reservoir. We also used serotyping to evaluate the clonal characteristics of our isolates and their distribution during the corresponding analysis.

Regarding the serotyping, following paragraphs has been added to the Introduction:

Introduction, Lines 69-73: “Among the available tools, serotyping, particularly of the O-antigen, is a classical and widely used method to differentiate E. coli lineages. Specific O-serotypes are associated with particular disease conditions and epidemiology, enabling tracking, and identification of high-risk lineages [18,19]. Although camel-associated E. coli is largely unexplored, serotyping allows comparison with ExPEC from other hosts and may reveal potential reservoirs of infection.”

4. L: 199-200 I don't understand this information: "Desvaux et al., [23]) was assessed in our samples of camel pathogenic E. coli."

This statement reflects that the 35 virulence-associated genes (VAGs) screened by PCR included those typical for intestinal pathogens (e.g., ETEC, STEC) as well as others related to adherence, invasion, iron acquisition, and serum resistance. Since characterizing ExPEC based on a few VAGs is challenging, we selected 17 prominent ExPEC VAGs for detection based on two key reviews (Sarowska et al. 2019 and Desvaux et al. 2020) and assessed their presence to determine the ExPEC potential of our samples.

For clarification, the following statement has been added:

Methods, Lines 168-173: “To assess the ExPEC potential of camel pathogenic E. coli isolates, we analyzed a set of 17 VAGs commonly linked to ExPEC strains (based on the reviews by Sarowska et al. [24] and Desvaux et al. [8]). These VAGs included: α-hly, afaI, cdt, cnf1, fimA, fyuA, ibeA, iroN, iss, iucC, ompT, pap, sat, sfa, sitA, traT, and tsh. For each isolate, an ExPEC-VAG score was calculated as the total number of detected ExPEC-specific VAGs. This approach allowed us to quantify the ExPEC potential.”

Lines 225-226: “…was assessed in our samples of camel pathogenic E. coli to evaluate their ExPEC potential (see Methods).”

5. The discussion on virulence factors should be more specific. What are the consequences of the presence of these genes and what role could they play in pathogenesis? Only statistics are given without indicating which Vf are important.

The detected VAGs are discussed in a greater detail and incl

---

## [Decision Letter · Decision Letter 1]

23 Sep 2025

Extraintestinal Escherichia coli from camel carcasses: Phylogroups, serotypes, and markers of virulence

PONE-D-25-05895R1

Dear Dr. Bosák,

We’re pleased to inform you that your manuscript has been judged scientifically suitable for publication and will be formally accepted for publication once it meets all outstanding technical requirements.

Kind regards,

Nabi Jomehzadeh, Ph.D (Associate Professor)

Academic Editor

PLOS ONE

Additional Editor Comments (optional):

Reviewer #1:

Reviewer #2:

Reviewers' comments:

Reviewer's Responses to Questions

**Comments to the Author**

Reviewer #1: All comments have been addressed

Reviewer #2: All comments have been addressed

2. Is the manuscript technically sound, and do the data support the conclusions?

Reviewer #1: (No Response)

Reviewer #2: Yes

3. Has the statistical analysis been performed appropriately and rigorously?

Reviewer #1: (No Response)

Reviewer #2: Yes

4. Have the authors made all data underlying the findings in their manuscript fully available?

Reviewer #1: (No Response)

Reviewer #2: Yes

5. Is the manuscript presented in an intelligible fashion and written in standard English?

Reviewer #1: (No Response)

Reviewer #2: Yes

Reviewer #1: (No Response)

Reviewer #2: I suggest that the conclusion should go beyond summarizing results. I recommend emphasizing the study’s contribution to the field, limitations, and directions for future research to align with the expectations of high-impact journals.

I agree to the acceptance of the article in this journal.

**Do you want your identity to be public for this peer review?** For information about this choice, including consent withdrawal, please see our Privacy Policy

Reviewer #1: No

Reviewer #2: No

---

## [Editor Report · Acceptance letter]

PONE-D-25-05895R1

PLOS ONE

Dear Dr. Bosák,

I'm pleased to inform you that your manuscript has been deemed suitable for publication in PLOS ONE. Congratulations! Your manuscript is now being handed over to our production team.

Kind regards,

on behalf of

Dr. Nabi Jomehzadeh

Academic Editor

PLOS ONE